# A Novel Lipid Metabolism and Endoplasmic Reticulum Stress-Related Risk Model for Predicting Immune Infiltration and Prognosis in Colorectal Cancer

**DOI:** 10.3390/ijms241813854

**Published:** 2023-09-08

**Authors:** Haoran Jin, Bihan Xia, Jin Wang, Shaochong Qi, Weina Jing, Kai Deng, Jinlin Yang

**Affiliations:** 1Department of Gastroenterology and Hepatology, West China Hospital, Sichuan University, Chengdu 610017, China; jin_haoran98@163.com (H.J.); xbh0715@163.com (B.X.); wangjin8240@wchscu.cn (J.W.); victorqiqsc@163.com (S.Q.); daisyjingwn@163.com (W.J.); 2Sichuan University-University of Oxford Huaxi Joint Centre for Gastrointestinal Cancer, Frontiers Science Center for Disease-Related Molecular Network, West China Hospital, Sichuan University, Chengdu 610017, China

**Keywords:** colorectal cancer, lipid metabolism, endoplasmic reticulum stress, prognostic prediction model, immune infiltration, drug sensitivity, immunotherapy

## Abstract

Lipid metabolism and endoplasmic reticulum stress exhibit crosstalk in various cancer types, which are closely associated with the progression of colorectal cancer (CRC). This study constructs a prognostic signature based on lipid metabolism and endoplasmic reticulum stress-related genes (LERGs) for CRC patients, aiming to predict the prognosis and immune response. RNA sequencing and clinical data from the TCGA and GEO databases were analyzed to identify differentially expressed LERGs with prognostic relevance using univariate Cox regression. Subsequently, a risk model was developed using the LASSO regression. CRC patients were stratified into low-risk and high-risk groups based on risk scores, with the high-risk cohort demonstrating a poorer clinical prognosis in multiple databases. The risk model showed robust correlations with clinical features, gene mutations, and treatment sensitivity. Significant differences in immune cell infiltration and the expression of immune-related factors were also detected between risk groups, and elevated scores of cytokines and failure factors were detected in single-cell RNA sequencing analysis. This research indicates that lipid metabolism and endoplasmic reticulum stress in CRC are correlated with tumor progression, an immunosuppressive landscape, and alterations of drug sensitivity. The developed risk model can serve as a powerful prognostic tool, offering critical insights for refining clinical management and optimizing treatment in CRC patients.

## 1. Introduction

Colorectal cancer (CRC) has become the third-most common cancer and the second-most frequent cause of cancer-related deaths [1]. Unfortunately, about 20% of CRC cases have metastases at diagnosis [2]. Clinical outcomes for CRC patients have markedly enhanced due to progress in surgical methods, antitumor medications, and other superior treatment strategies [3]. Compared to traditional chemotherapy and targeted therapy, immunotherapy has several advantages, including durable immune responses, fewer side effects, and extended progression-free survival for metastatic CRC patients with DNA mismatch repair deficiency or microsatellite instability [4,5]. However, some of these patients enter into the stage of immune resistance quickly [6]. Furthermore, CRC exhibits heterogeneity across multiple levels, ranging from epigenetic and genetic driver events to signaling pathway reconfiguration, as evidenced by differential gene expression patterns [7]. In addition, the dynamic remodeling of the tumor microenvironment (TME) infiltrated with various stromal and immune cells makes CRC spatially heterogeneous [8]. This complexity in molecular biology and TME components significantly influences CRC hallmarks, the tumor response to therapies, and patient outcomes [7]. Therefore, it is crucial to identify subsets of CRC based on molecular signatures in order to accurately predict prognosis and enable precisely targeted interventions.

Aberrant lipid metabolism is emergingly recognized as a hallmark feature of many cancers, including CRC [9]. Lipids, diverse in their classification, include phospholipids and cholesterol, fundamental components of the lipid bilayers of cellular membranes. Triglycerides, the most common type of lipid, serve as the primary form of energy storage, while fatty acids (FAs) are a significant energy source derived from triglyceride breakdown. Lipid-related factors, such as obesity, dyslipidemia, and a high dietary fat intake, have been associated with an increased risk of developing CRC [10,11,12]. Furthermore, the dysregulation of lipid metabolism has been experimentally implicated in CRC initiation, progression, and chemoresistance [13,14,15]. Cancer cells undergo different stages that necessitate lipid-related metabolic and structural adaptations, such as altering the lipid membrane composition and promoting lipid catabolism and anabolism, to trigger intracellular stress protection (e.g., oxidative stress and ER stress) and evade programmed cell death pathways (e.g., apoptosis, pyroptosis, and ferroptosis). On the other hand, cancer cells remodel lipid metabolism to interact with neighboring stromal and immune cells and reshape the TME to resist therapy and promote relapse [16]. Thus, identifying the lipid metabolism-dependent molecular mechanisms and implementing targeted interventions may offer therapeutic opportunities to prevent CRC progression.

The endoplasmic reticulum (ER) is an essential cellular organelle involved in protein synthesis and folding, lipid biosynthesis and metabolism, and calcium storage and signaling to determine cell fate and survival [17]. Various intrinsic and extrinsic factors can disturb ER homeostasis. Genetic, transcriptional, and metabolic abnormalities, as well as unfavorable microenvironmental conditions due to nutrient deprivation, oxygen limitation, and heightened metabolic demands within tumors, can induce a state of “ER stress”, marked by an accumulation of misfolded or unfolded proteins. ER stress instigates the unfolded protein response (UPR), which strives to re-establish ER stability and facilitate cellular adaptation within the TME [18,19]. Persistent or severe ER stress has been demonstrated to govern multiple pro-tumoral attributes in cancer cells and immune cells, leading to diverse effects ranging from cellular reprogramming and adaptation to autophagy and apoptosis [18]. Furthermore, the capacity to tolerate this persistent ER stress promotes cancer cell survival, angiogenesis, metastasis, chemoresistance, and immunosuppression [17]. In recent years, ER stress-related signaling pathways have emerged as critical regulators of tumor growth; metastasis; and the response to chemotherapy, targeted therapy, and immunotherapy [19]. Spatial transcriptomics data revealed that ER stress-related gene patterns in CRC were involved in forming the immunosuppressive TME and predicting patient prognosis [20].

Emerging evidence suggests a reciprocal relationship between dysregulated lipid metabolism and ER stress in many malignant diseases. Dysregulated lipid metabolism can contribute to ER stress through lipid imbalances, oxidative stress, and altered protein translation [21,22]. ER stress, in turn, dysregulates lipid metabolism through activating signaling pathways and the UPR-mediated transcriptional and post-translational regulation of lipid-related genes [23,24]. Therefore, exploring the role of lipid metabolism and ER stress-related genes (LERGs) in CRC might help us to understand their interplay and develop new treatment strategies. In this study, differential LERGs in normal versus tumor groups were selected to construct a risk model for predicting the prognosis of CRC. Simultaneously, the risk model demonstrated commendable effectiveness in forecasting the clinical attributes, immune environment, and drug responsiveness across numerous bulk RNA-seq and single-cell RNA sequencing (scRNA-seq) cohorts. Low-risk patients experienced significantly extended survival times compared to high-risk patients, suggesting that our signature could assist in differentiating survival situations in CRC, potentially offering a guideline for precise intervention in clinical applications.

## 2. Results

### 2.1. Exploration of Differentially Expressed LERGs in CRC

Differential genes were identified using the “limma” R package, with the selection criteria being a *p*-value < 0.05 and |FoldChange| > 1.5 (|log_2_FoldChange| > 0.585) [25]. This resulted in the identification of 4940 key genes exhibiting differential expression during the development of CRC, comprising 2325 up-regulated genes and 2615 down-regulated genes (Figure 1A) (Appendix A). Genes from the ER stress and lipid metabolism gene sets with a relevance score > 5 were extracted and intersected with the differential genes, yielding 271 intersecting genes (Figure 1B). Subsequently, the Cox Proportional-Hazards Model identified 28 prognostically significant genes (*p*-value < 0.05) (Figure 1C) (Appendix A).

### 2.2. Construction of the Prognostic Model 

To further identify key genes within the prognostic gene set, we collected clinical information from CRC patients and utilized the lasso regression to screen for characteristic genes (Figure 1D–F) (Appendix A). Patients in TCGA were randomly assigned to a training dataset and a test dataset at a ratio of 4:1. Following the LASSO regression analysis, a risk score was generated based on 18 prognostically significant genes (Risk Score = MMP3 × (−0.097771129) + HSPA8 × (−0.06324672) + PTGIS × (−0.056664141) + CPT2 × (−0.056371262) + GPI × (−0.031097946) + CDKN2A × 0.011962818 + SPP1 × 0.024967708 + GRP × 0.025154411 + APOE × 0.032478659 + CHGA × 0.032667622 + CAV1 × 0.040915171 + GSTM1 × 0.05604283 + VEGFA × 0.073841708 + LAMA2 × 0.083911131 + TIMP1 × 0.10382603 + AGRN × 0.107521782 + HSPA1A × 0.186021817 + SNAP25 × 0.224930455). This risk model is used to predict the levels of ER stress and lipid metabolism within the tumor. Based on the median risk score for each sample, patients were divided into high- and low-risk groups, with survival outcomes analyzed using Kaplan–Meier curves. In both the TCGA training dataset (*p* < 0.001) and test dataset (*p* = 0.016), the overall survival (OS) of the high-risk group was significantly lower than that of the low-risk group. Furthermore, the ROC curve results for both the TCGA training and test datasets indicated the model’s good predictive performance (Figure 2A,B).

### 2.3. Validation of the Prognostic Model across Multiple External Datasets

Data of survival information (GSE12945, GSE17536) from the GEO database for CRC patients were downloaded. Patients in validation datasets were assigned to high- and low-risk groups for survival analysis according to their median risk scores. The results showed that the OS of the high-risk group was significantly lower than that of the low-risk group in the external validation datasets GSE12945 (*p* = 0.002) and GSE17536 (*p* = 0.042). To verify the model’s accuracy, we performed ROC curve analysis on the model using external datasets. The results showed that the model has a strong predictive performance and stability for CRC patient prognosis (Figure 2C,D).

### 2.4. Incidence Risk and Independent Prognostic Analysis

Via univariate analysis (HR = 3.628 (2.687–4.898), *p* < 0.001) and multivariate analysis (HR = 3.006 (2.129–4.244), *p* < 0.001), it was discovered that the risk score serves as an independent prognostic factor for CRC patients (Figure 3A,B) (Appendix A). The results were presented in a nomogram (Figure 3C). Additionally, predictive analyses were conducted for both 3- and 5-year periods for CRC, revealing that the nomogram possesses strong validation performance and predictive accuracy (Figure 3D–F).

### 2.5. Correlation Analysis of the Disease Risk Score with Multiple Clinical Indicators

Based on the clinical indicators, sample-associated risk score values were assigned to various groups. Upon employing the Wilcoxon test, it was observed that the distribution of risk score values showed significant inter-group differences within clinical indicators such as Stage, T, M, and N (*p* < 0.05) (Figure 4). These findings illustrated that the constructed risk-scoring model possesses good clinical applicability for indicating the classifications of CRC samples.

### 2.6. The Mutational Landscape for CRC and Tumor Mutational Burden (TMB)

The mutational landscape of patients within high- and low-risk groups was further explored. The findings indicated that patients in the high-risk group exhibited a significantly lower mutation ratio for genes such as APC compared to the low-risk group (76% vs. 82%), while the mutation ratios of TP53 and APC genes appeared to be higher than those in the low-risk group (Figure 5A,B). Subsequently, survival analysis was performed, correlating TMB and microsatellite instability (MSI) with risk scores, with the results depicted in the corresponding figures (Figure 5C,D).

### 2.7. Drug Sensitivity and Molecular Pathway for the Risk Model

Surgical treatment combined with chemotherapy has been proven to be effective for early-stage CRC [26]. Our findings indicated a significant correlation between the risk score and the sensitivity to 16 certain chemotherapy drugs. Patients in the low-risk group exhibited a higher sensitivity to Mitomycin C, Gemcitabine, AKT inhibitor VIII, and Tipifarnib. On the other hand, patients in the high-risk group showed a higher sensitivity to Docetaxel, Gefitinib, Pazopanib, and Rapamycin (Figure 6A). Subsequent analysis focused on the specific signaling pathways to explore the potential molecular mechanisms of how the risk score impacts tumor progression. GSVA analysis revealed that the main divergent pathways between the two patient cohorts were predominantly found in MYOGENESIS, EPITHELIAL MESENCHYMAL TRANSITION, and APICAL JUNCTION signaling pathways (Figure 6B). GSEA analysis disclosed that the PI3K−Akt and Rap1 signaling pathways were significantly enriched in the high-risk group, while nucleotide metabolism pathways were enriched in the low-risk group (Figure 6C).

### 2.8. Relationship between the Risk Model and the Immune Microenvironment

The TME primarily comprises cancer-associated fibroblasts, immune cells, an extracellular matrix, and various signaling, molecular, and cancer cells, which significantly influence the tumor diagnosis, outcomes, and treatment sensitivity [27]. This study further investigated the relationship between the risk model and immune infiltration in TME. Compared to immune infiltration in various risk groups, this study revealed a significant decrease in T cells CD4^+^ memory resting and T cells CD4^+^ memory activated in the high-risk group. Conversely, a significant increase was observed for naive B cells, regulatory T cells (Tregs), and M0 macrophages (Figure 7A). The correlation analysis between risk scores and immune cell contents emphasized a substantial positive correlation with M0 macrophages and naive B cells, alongside a negative correlation with resting and activated memory CD4^+^ T cells (Figure 7B). Moreover, the correlation between lipid metabolism/ER stress genesets and immune infiltration was illustrated via ssGSEA. M2 macrophages and naive B cells show a positive correlation with the levels of lipid metabolism/ER stress in tumor samples. Conversely, NK cells and CD8^+^ T cells show a negative correlation (Figure 7D). This suggests that under elevated levels of lipid metabolism/ER stress, the TME predominantly displays an immunosuppressive phenotype, which aligns with the immune infiltration analysis of our risk model. Further analysis was conducted on immune regulatory genes, and the expression differences of immune-related inhibitory factors and cytokines between high- and low-risk groups are represented (Figure 7E,F). Finally, a prediction of the sensitivity to tumor immunotherapy was carried out, indicating poorer responses to immunotherapy in the high-risk group (Figure 7C).

### 2.9. Construction of a WGCNA Co-Expression Network and Upstream Mechanism Analysis of Hub Genes

To further pinpoint key genes associated with colorectal cancer, a WGCNA was employed, based on the colorectal cancer transcriptome, to elucidate related co-expression regulatory networks [28]. A soft threshold β was set to six (Figure 8A,B), and a total of eight gene modules were identified via TOM (Figure 8C,D) (Appendix A). These comprised the black (99), blue (1052), brown (610), green (454), grey (569), red (176), turquoise (1572), and yellow (468) modules. The brown module exhibited the highest correlation (cor = 0.49, *p* = (5 × 10^−37^)). Intersecting genes from the brown module and the 18 model genes, 6 genes were delineated as key candidates for further study (Figure 8E), namely, MMP3, SPP1, APOE, CAV1, TIMP1, and AGRN. Subsequent PPI network analysis illustrated the interactions amongst these key genes (Figure 8F). An upstream transcription factor (TF) analysis of the key genes was also performed. The results showed that the MOTIF annotation cisbp_M5419 had an NES of 4.9, and three key genes were enriched in this motif. The top six motifs and their corresponding TFs associated with the hub genes are presented in Figure 8G.

### 2.10. Validation of Hub Genes in the CRC Clinical Sample

We finally explored the gene expression using 12 pairs of clinical samples in quantitative real-time polymerase chain reaction qRT-PCR and pathology slide images from the HPA database (Figure 9A–E). Among these six hub genes, AGRN, SPP1, MMP3, and TIMP1 exhibited significantly elevated expressions in CRC tissues compared with normal tissues. Additionally, CAV1 exhibited significantly decreased expression in both RNA and protein levels. Although APOE did not show significant changes in qRT-PCR, the protein levels showed decreased trends, consistent with the TCGA analysis.

### 2.11. Single-Cell Analysis for the Immune Landscape and Gene Expression in CRC 

Cell clustering was achieved using the t-SNE algorithm, resulting in the identification of 17 subtypes (Figure 10A). The R package “SingleR” was utilized to annotate each subtype, with the 17 clusters falling within 9 cellular categories: T cells, B cells, epithelial cells, monocytes, neutrophils, tissue stem cells, macrophages, NK cells, and endothelial cells (Figure 10B). The expression of the six hub genes across these nine cell types was visually represented (Figure 10G). It was discerned that intricate interaction patterns exist between these cellular subtypes in the high-risk and low-risk groups. Macrophages were found to have the most potential interactions with other cells in both risk groups (Figure 10C,D). Following this, failure scores and cytokine scores were obtained using the GeneCards database and were compared between the high-risk and low-risk groups. The results of this analysis were depicted (Figure 10E,F).

## 3. Discussion

The interplay between dysregulated lipid metabolism and ER stress has been identified in many cancer types. The highly expressed OGR1 mediates lipogenesis, enabling metabolic adaptation to acid stress in breast cancer cells. This adaptation is achieved through the activation of ER stress and autophagy [22,29]. Interrupting mitochondrial lipid oxidation via CPT1 results in a transient surge in glucose uptake, ultimately culminating in ER stress and apoptosis in prostate cancer cells [30]. Several studies have highlighted the significance of crosstalk between dysregulated lipid metabolism and ER stress in CRC. The expression of fatty acid synthase (FASN), the cellular enzyme synthesizing palmitate, was associated with CRC patient survival [31]. The inhibition of FASN induced ER stress-mediated cell death and even enhanced the antitumor effect of 5-FU by causing lipid disequilibrium [32,33]. Cholesterol in the colorectal TME could induce the expression of immune checkpoints and the exhaustion of CD8^+^ T cells by activating the ER stress sensor XBP1 and regulating PD-1 transcription. Thus, cholesterol deprivation could serve as a novel approach to restoring T cell-based immunotherapy [34]. In addition, a high-fat diet was reported to aggravate colitis-associated carcinogenesis by evading ER stress-related ferroptosis, a type of programmed cell death caused by lipid peroxidation [35]. Therefore, constructing a risk model based on a combination of both dysregulated lipid metabolism and ER stress for predicting CRC patient survival and immune landscapes may hold substantial significance and value for clinical applications.

In this study, 18 differential expression genes related to lipid metabolism and ER stress were identified to construct a risk model for predicting survival outcomes. A higher risk score indicates a stronger relationship with lipid metabolism and ER stress within the tumor. Furthermore, the risk model based on LERGs has been proven to have significant efficacy in prognostic prediction in TCGA and two external validation datasets. The AUC values at 1, 3, and 5 years were 0.7, 0.75, and 0.74, respectively, in the TCGA testing dataset and 0.6, 0.63, and 0.64, respectively, in the GSE12945 and GSE17536 datasets. These results suggest that the performance of the risk model is reliable. Several LERGs have been proven to be associated with the development and progression of CRC. For example, GRP-derived peptides are expressed in CRC cell lines and tumors and are biologically active in vivo, which stimulates the proliferation of CRC cells and normal colonic mucosa [36]. LAMA2 has been identified as a sphingolipid-metabolism-related gene predicting prognosis and immune infiltration in CRC [37]. The overexpression of PTGIS has been linked with poor prognosis and the potential prediction of liver metastasis in CRC patients [38]. In addition, the downregulation of CPT2, a key enzyme of FAO, promotes proliferation and inhibits apoptosis through the p53 pathway in CRC [39]. Moreover, GSM1 gene mutation is associated with an increased risk of CRC [40]. These findings highlight the significance of these genes in CRC and their potential as prognostic markers or therapeutic targets. Furthermore, we used the WGCNA algorithm to identify crucial LERGs for experimental verification. As a result, six hub genes (AGRN, APOE, CAV1, SPP1, MMP3, and TIMP1) were identified to further explore the mechanism of CRC progression. Extensive research has demonstrated a significant association between these hub genes and lipid metabolism and ER stress [41,42]. Particularly, the significant role of CAV1 was emphasized [43,44]. The expression of APOE and SPP1 has been detected in macrophages during tumor progression [45], with SPP1 becoming highly expressed in lipid-associated macrophages [46]. Further, MMP3, TIMP1, and AGRN have been associated with CRC progression [47,48,49]. However, the exact mechanisms by which these hub genes contribute to CRC progression via lipid metabolism and ER stress have yet to be fully elucidated. Preliminary transcription factor predictions propose that the ETS1 and EGR families may exert potential regulatory effects.

Our prognostic model holds significant clinical value in both prognostic prediction and drug therapy. Patients in the high-risk group were found to have a poorer prognosis and a higher TNM stage compared to those in the low-risk group. The risk score was identified as an independent predictive element for CRC patients, as demonstrated by both univariate and multivariate Cox regression analyses. Moreover, a nomogram was constructed using risk scores and additional clinical features. The calibration and DCA curves show high predictive accuracy and clinical applicability. To further investigate the differential molecular mechanisms between the two risk groups, we employed GSVA and GSEA. The pathways of myogenesis, epithelial–mesenchymal transition, apical junction, PI3K-Akt, and Rap1 signaling were notably enriched in the high-risk group. These pathways play crucial roles in tumoral cell growth, adhesion, and metastasis [50,51,52]. Conversely, pathways such as MYC targets, oxidative phosphorylation, and nucleotide metabolism were significantly enriched in the low-risk group, suggesting the presence of metabolic alteration between the two subgroups. Subsequent investigations have revealed differential drug sensitivities across high- and low-risk groups, implying potential heterogeneity in therapeutic responses. Gemcitabine, AKT inhibitor VIII, and Tipifarnib have already been recognized as effective treatments for CRC [53,54,55]. However, drug sensitivity analysis reveals that the high-risk group exhibits elevated IC50 values for these drugs, suggesting a potential reduced sensitivity in high-risk patients. For high-risk group patients, drugs with a higher sensitivity, such as Docetaxel, Gefitinib, and Pazopanib, can be considered as preferable therapeutic strategies for the precision treatment of CRC.

Furthermore, the immune cell infiltration was assessed based on the TCGA samples. The analysis revealed an increase in M0 macrophages and naive B cells, as well as a decrease in CD4^+^ memory T cells and dendritic cells within the high-risk group. This pattern suggests an overall trend toward immune suppression, potentially contributing to a poorer prognosis. Additionally, the high-risk group exhibited an elevated expression of immune-inhibitory factors and a decreasing trend in the expression of immune-stimulatory factors, providing further evidence of an immunosuppressive environment and poor immunotherapy response. ScRNA-seq analysis has indicated strong macrophage communication in the high-risk group. Meanwhile, a pivotal study highlights the significant role of activated endoplasmic reticulum stress and lipid metabolism in tumors for enhancing the polarization of macrophages in the TME, thereby promoting tumorigenic functionality [56,57]. This finding aligns with the elevated macrophage expression observed in the high-risk group in our research. Notably, the hub genes, SPP1 and APOE, are significantly enriched in macrophages. Elevated scores of cytokines and failure factors were also detected, which is related to tumor progression, abnormal activation of the immune system, and increased inflammatory responses.

Although our findings have potential clinical significance, there are still some limitations. First and foremost, our analysis and conclusions were based on tumor samples collected from the TCGA database, and the validity of our findings was corroborated using multiple cohorts from the GEO database. However, a more comprehensive validation would necessitate the collection of clinical samples from a significantly larger, independent prospective cohort to further verify the risk model we established in this study. Second, the sample size for our verification experiments was limited, restricting us to only validating the transcriptional expression levels of the hub genes. To further explore the role of LERGs in CRC progression, more exhaustive in vivo and in vitro experiments are indispensable. Despite these shortcomings, it is undeniable that our study represents the first investigation into the influence of lipid metabolism and ER stress-related genes on CRC progression, which provides a robust theoretical foundation for in-depth mechanistic exploration. Meanwhile, we specifically focused on predicting clinical outcomes, the tumor immune microenvironment, and drug treatment strategies. This pioneering research offers potentially promising clinical implications.

## 4. Materials and Methods

### 4.1. Data Sources and Collection

Transcriptomic analyses of 51 normal and 650 CRC samples were retrieved from The Cancer Genome Atlas (TCGA) database (https://portal.gdc.cancer.gov/, accessed on 10 May 2023). Additionally, clinical data from 630 cases and SNP data from 551 cases (Appendix A) were obtained. Differentially expressed genes were identified using the R package “limma”. The GSE12945 and GSE17536 Series Matrix File data were downloaded from the Gene Expression Omnibus (GEO) database (https://www.ncbi.nlm.nih.gov/geo/, accessed on 10 May 2023). Data from 62 cases and 177 cases of CRC patients from 2 datasets were retained for subsequent validation analysis after excluding samples with incomplete expression profiles and survival information (Appendix A). In addition, the GSE188711 Series Matrix File from six CRC patient cases was extracted for scRNA-seq analysis. Moreover, ER stress (Appendix A) and lipid metabolism (Appendix A) genes were obtained from the GeneCards database (https://www.genecards.org/, accessed on 10 May 2023).

### 4.2. Construction and Validation of a Risk Model

To confirm the underlying prognosis-related genes, we performed univariate Cox regression for all differential expressed lipid metabolism and ER stress-related genes in TCGA cohorts. Then, the LASSO regression method was performed to obtain the best LERGs and their regression coefficients.
Riskscore = ∑i Coefficient (i) × Expression of gene (i)

CRC patients in the TCGA, GSE12945, and GSE17536 datasets were divided into low- and high-risk categories using the median risk score. The high-risk and low-risk groups were compared using the survival analysis, while the model’s accuracy was evaluated via ROC curves.

### 4.3. Construction of a Predictive Nomogram

By integrating multiple risk elements, the nomogram is a great tool for quantifying the risk for individuals. Using the R package “rms”, we constructed a nomogram that assigns scores to each level of influence factor based on their contribution to the outcome variable.

### 4.4. TMB Analysis

TMB, representing the number of coding somatic indel mutations and base substitutions per megabase, was calculated during the genome analysis. For this study, the R package “maftools” was utilized to detect mutations in the model [58].

### 4.5. Drug Sensitivity Analysis

IC50 represents the half-maximal inhibitory concentration, denoting the drug concentration required to achieve 50% inhibition in a cell line. Based on the Genomics of Drug Sensitivity in Cancer (GDSC) database (https://www.cancerrxgene.org/, accessed on 10 May 2023), which is the largest pharmacogenomics database, we used the R package “pRRophetic” to predict the chemotherapy sensitivity of tumor samples [59]. A Wilcoxon signed-rank test was applied to compare the IC50 of different therapeutic drugs in high-risk and low-risk groups.

### 4.6. Gene Set Variation Analysis (GSVA) and Enrichment Analysis (GSEA)

GSVA, an unsupervised and non-parametric approach, is utilized to assess the enrichment of gene sets in correlation with mRNA expression in each sample [60]. Gene sets downloaded from the MSigDB (https://www.gseamsigdb.org/gsea/msigdb/, accessed on 10 May 2023) database were comprehensively scored using the GSVA algorithm. 

GSEA was performed to compare the differences in the gene set between the high- and low-groups, with cutoff values of *p* < 0.05 and a normalized enrichment score (NES) > 1 [61]. It showed enriched pathways to evaluate potential changes in the biological function between the two subgroups.

### 4.7. Immune Infiltration Analysis

CIBERSORT is a method extensively utilized for assessing immune cell types within the microenvironment [62]. Based on the principles of support vector regression, it deconvolutes the expression matrices of immune cell subtypes. It comprises 547 biomarkers and distinguishes 22 human immune cell phenotypes, including T cells, B cells, plasma cells, and myeloid subgroups. In this study, the CIBERSORT algorithm was applied to patient data to infer the relative proportions of the 22 types of immune infiltrating cells (Appendix A). A Spearman correlation analysis was conducted on gene expression levels and the immune cell content. Furthermore, to quantify the correlation between lipid metabolism/ER stress and immune cell infiltration in each sample, single-sample Gene Set Enrichment Analysis (ssGSEA) was used to assess the enrichment of immune cells in the tumor samples. The Tumor Immune Dysfunction and Exclusion (TIDE) (http://tide.dfci.harvard.edu/, accessed on 10 May 2023) algorithm was used to compare the differences in immunotherapy efficacy between the high- and low-risk groups [63].

### 4.8. Weighted Gene Co-Expression Network Analysis (WGCNA) 

To identify core genes among the 18 model genes, we utilized the WGCNA algorithm. This approach generated a weighted gene co-expression network, enabling the recognition of co-expressed gene modules and the relationships between gene networks and phenotypes [64]. The R package “WGCNA” was used to construct a co-expression network for all genes, selecting the top 5000 genes by variance for further scrutiny, with a soft-thresholding power of six. The weighted adjacency matrix was converted into a Topological Overlap Matrix (TOM), which was used for network connectivity estimation and hierarchical clustering. This process led to identifying distinct genes represented by different colors in a dendrogram, thus classifying thousands of genes into several modules based on their expression patterns.

### 4.9. Construction of a PPI Network and Prediction of the Transcription Factor (TF) 

The hub genes filtered via the WGCNA analysis were further investigated for their interactions and TF regulatory mechanisms. The protein–protein interaction (PPI) network was constructed using the STRING database (https://string-db.org/, accessed on 10 May 2023). TFs were predicted using the R package “RcisTarget”, with all computations based on motifs [65]. The Normalized Enrichment Score (NES) of a motif depends on the total number of motifs in the database. In addition to the annotations provided by the original data for base sequences, further annotation files were inferred based on base sequence similarity and gene sequencing.

### 4.10. Clinical Samples’ Verification of Hub Genes

Twelve pairs of CRC and adjacent normal tissues were obtained post-surgically from West China Hospital of Sichuan University, with ethical committee approval. The samples were preserved at −80 °C until required. The total RNA was extracted from these samples using TRIzol reagent (Invitrogen, Carlsbad, CA, USA) and subsequently transcribed into cDNA using the PrimeScript RT kit (Vazyme, Nanjing, China). The cDNA concentrations were assessed utilizing the SYBR qPCR Master Mix (Vazyme, Nanjing, China). The 2^−ΔΔCt^ method was employed to calculate the relative expression levels of signature genes, using U6 as the normalization control. Variations in the expression levels between healthy and tumor tissues were examined using both paired and unpaired *t*-tests. The sequences for the qRT-PCR primers can be found in Appendix A. Lastly, CRC tissue images illustrating hub gene expressions were retrieved from the Human Protein Atlas (HPA) database (https://www.proteinatlas.org/, accessed on 10 May 2023).

### 4.11. Single-Cell RNA Sequencing Dataset Analysis

The GSE188711 data were processed using the R package “Seurat”, and the position relationships between each cluster were analyzed using the t-SNE algorithm. The clusters were annotated using the R package “SingleR” and were assigned to various cells of significant relevance to tumor progression. The interaction relationship in the single-cell transcriptome features was analyzed separately in the high- and low-risk groups using the R package “CellChat” [66].

### 4.12. Statistical Analysis

Survival curves were generated using the Kaplan–Meier method and compared via a log-rank test. A multivariate analysis was carried out utilizing the Cox proportional hazards model. All statistical analyses were performed using R version 4.2.3, with *p*-values less than 0.05 considered statistically significant (* *p* < 0.05; ** *p* < 0.01; *** *p* < 0.001; and **** *p* < 0.0001).

## 5. Conclusions

In summary, we identified 18 differentially expressed genes related to lipid metabolism and ER stress, which were used to construct a risk model. Our risk model is associated with overall survival, the effectiveness of antitumor therapy, and immune infiltration in CRC patients. This reveals that the elevated levels of ER stress and lipid metabolism are correlated with an immunosuppressive landscape and poor prognosis in CRC, which provides strategic insights for precision therapy. By exploring new directions in lipid metabolism and ER stress, this study could aid in personalized treatment strategies and provides a significant foundation for further research on the immune microenvironment of CRC.

## Figures and Tables

**Figure 1 ijms-24-13854-f001:**
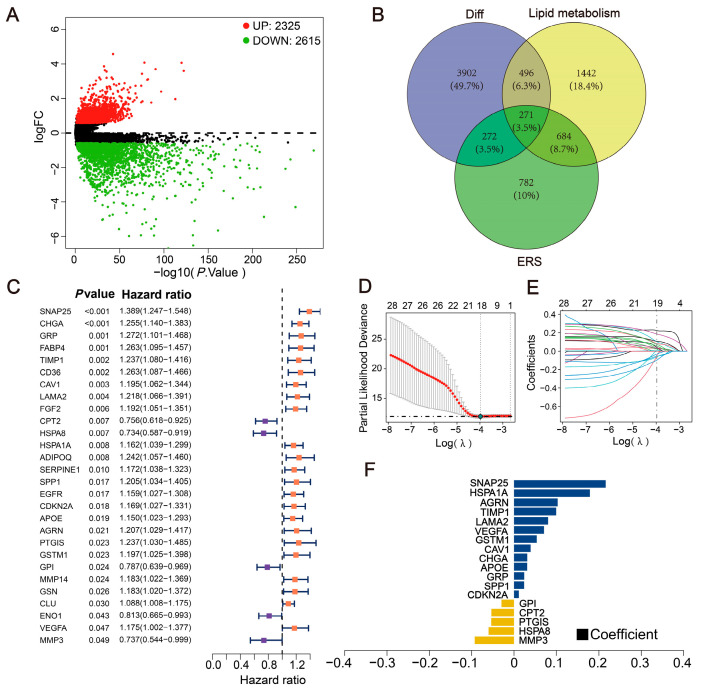
Construction of the prognostic signature based on differentially expressed LEGRs. (**A**) The volcano plot of differently expressed genes in TCGA. (**B**) The Venn diagrams screening differently expressed LERGs. (**C**) A total of 28 survival-related genes via Cox regression. (**D**,**E**) LASSO coefficient profiles of the 28 survival-related genes. (**F**) Coefficient profile plot produced against the log (lambda) sequence in the LASSO model. The optimal parameter (lambda) was selected as the first dotted black line indicated.

**Figure 2 ijms-24-13854-f002:**
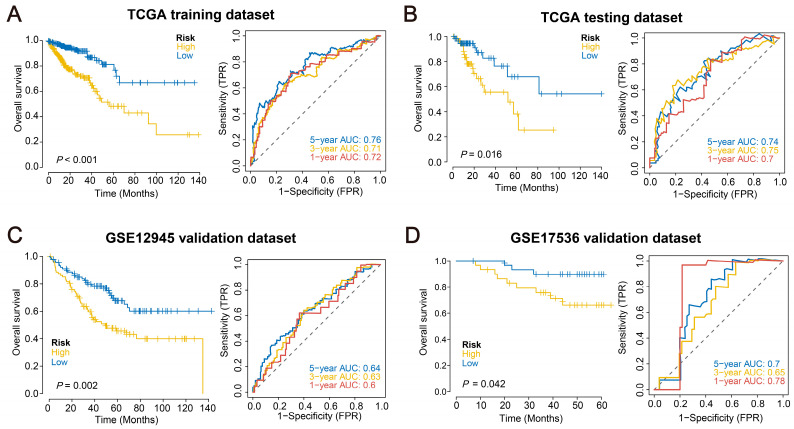
Evaluation of the risk model’s efficiency in predicting patient survival. (**A**–**D**) Survival curves for the high-risk and low-risk groups are exhibited, as well as the ROC curves and AUCs for predicting 1-year, 3-year, and 5-year OS in the TCGA training dataset, TCGA test dataset, GSE12945 validation dataset, and GSE17536 validation dataset.

**Figure 3 ijms-24-13854-f003:**
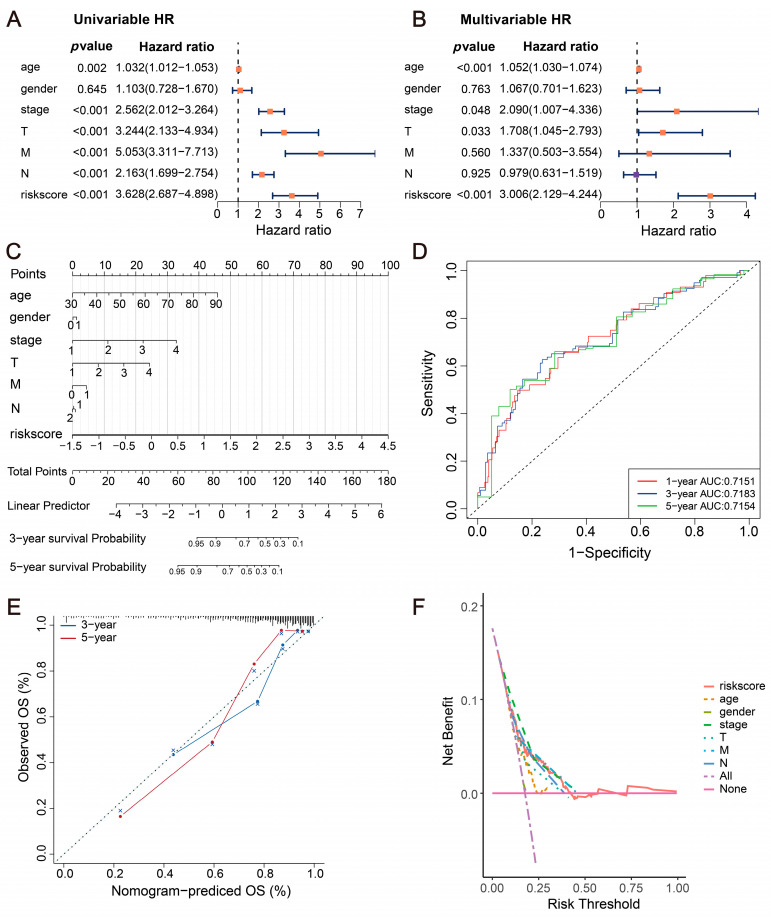
Construction of a predictive nomogram for evaluating patient survival. (**A**,**B**) Independent clinical prognostic factors for CRC patients identified via univariate and multivariate Cox regression. (**C**) A nomogram for predicting OS in CRC patients. (**D**) ROC curves. (**E**) Calibration plots for predicting 3-year and 5-year survival for patients. (**F**) DCA for evaluating clinical utility.

**Figure 4 ijms-24-13854-f004:**
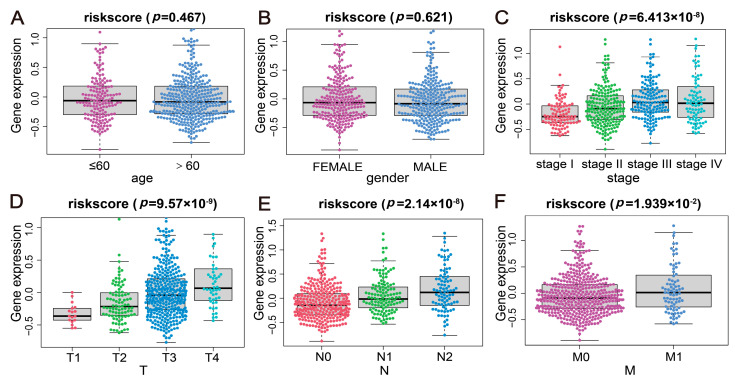
Correlation between the risk score and multiple clinical indicators. (**A**–**F**) Clinical indicators, including age, gender, stage, T, N, and M, were evaluated for their correlation with the risk score in boxplots.

**Figure 5 ijms-24-13854-f005:**
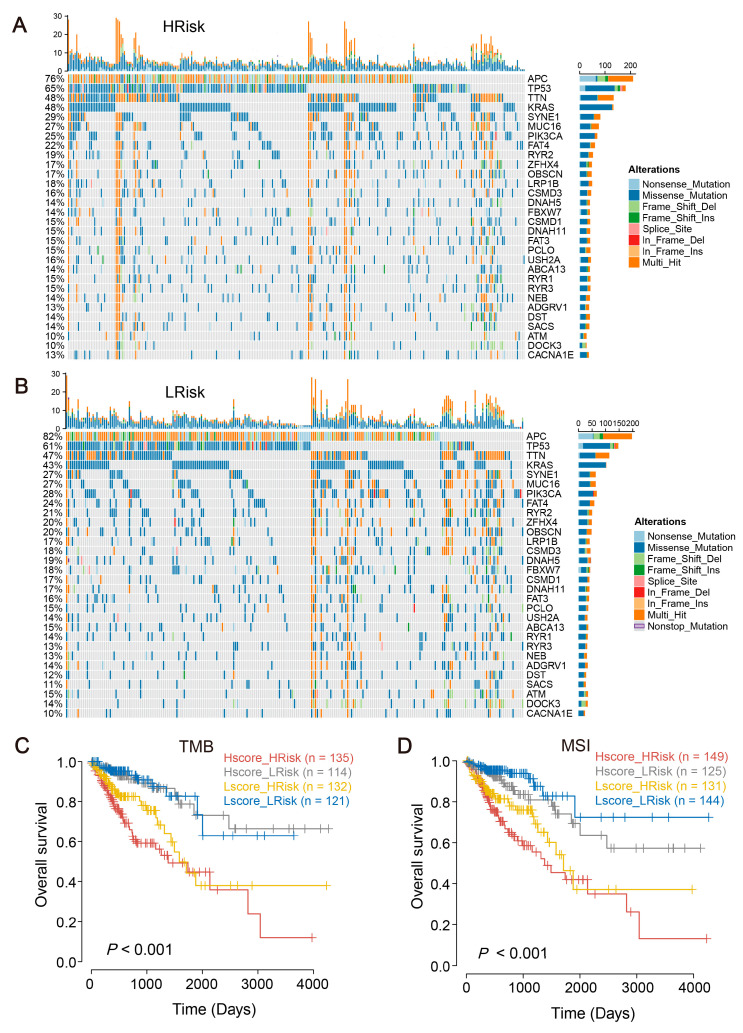
Schemes follow the same formatting. (**A**,**B**) The mutated genes of the high-risk group and the low-risk group from the TCGA cohort. (**C**) Combined TMB and risk score for predicting CRC patient survival. (**D**) Combined MSI and risk score for predicting CRC patient survival.

**Figure 6 ijms-24-13854-f006:**
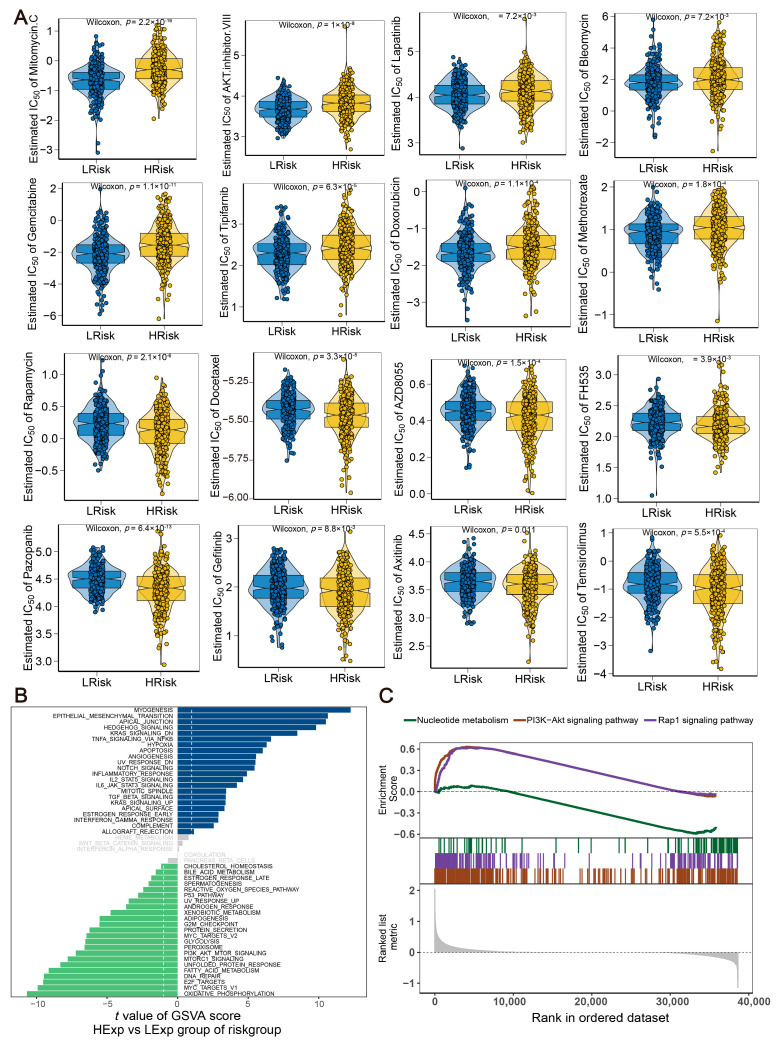
Drug sensitivity and molecular pathway for the risk model. (**A**) Therapeutic drugs showing significant IC50 differences in high- and low-risk groups. (**B**) GSVA results revealing the main divergent pathways. (**C**) GSEA results showing the top three pathways in two subgroups.

**Figure 7 ijms-24-13854-f007:**
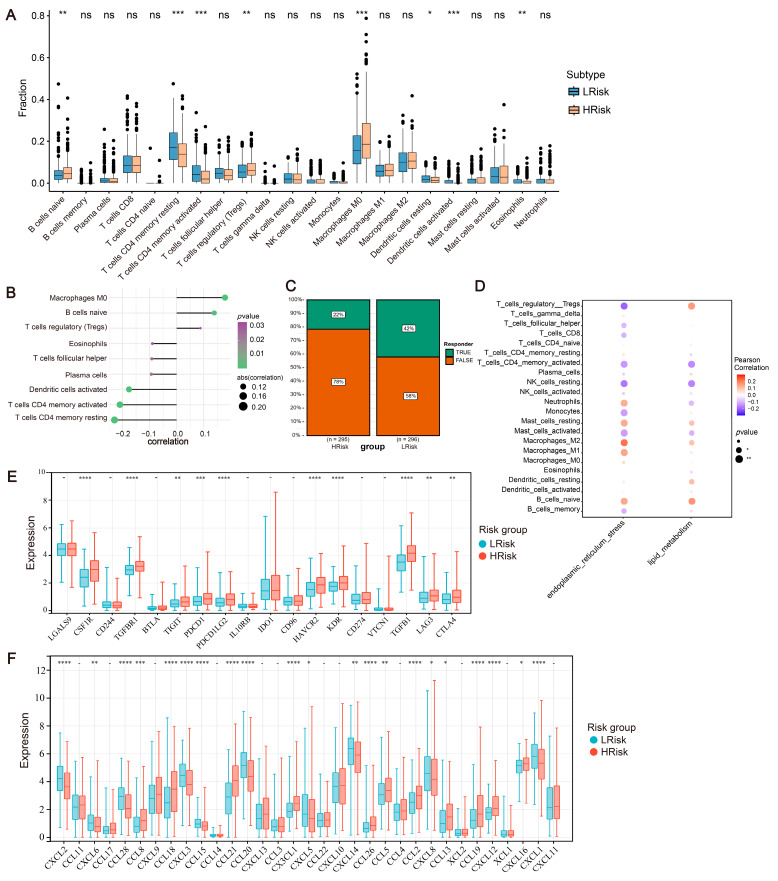
The different immune landscapes between the low- and high-risk groups. (**A**) CIBERSORT analysis. (**B**) Correlation between risk score and immune cell content. (**C**) Correlation between risk score and immunotherapy response. (**D**) Correlation between lipid metabolism/ER stress and immune infiltration via ssGSEA. (**E**) Expression variation in the immune checkpoint. (**F**) Expression variation of chemokines. * *p* < 0.05; ** *p* < 0.01; *** *p* < 0.001; and **** *p* < 0.0001. ns *p* > 0.05.

**Figure 8 ijms-24-13854-f008:**
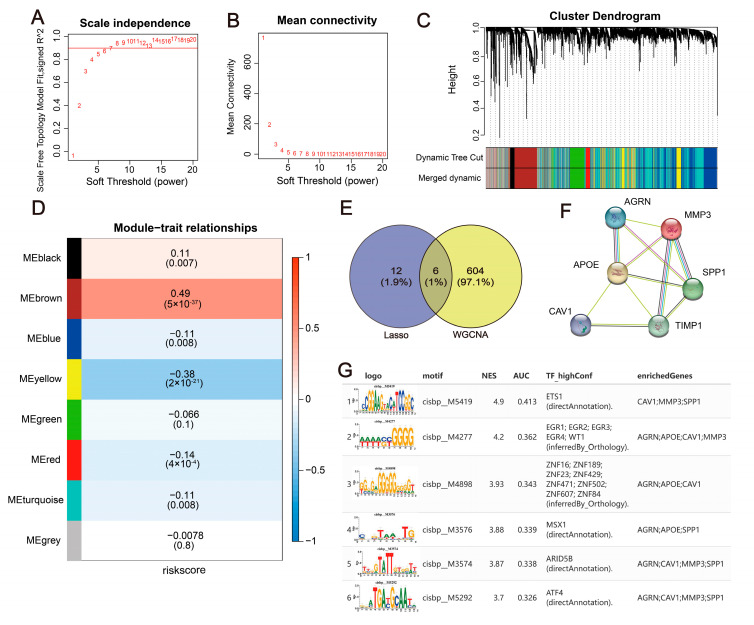
Construction of a WGCNA co-expression network and upstream mechanism analysis of hub genes. (**A**,**B**) A soft threshold β was set to six. (**C**) A total of eight modules were obtained. (**D**) The brown module is most related to the risk sore. (**E**) The Venn diagrams for screening hub genes. (**F**) PPI analysis in six hub genes. (**G**) TF prediction of hub genes.

**Figure 9 ijms-24-13854-f009:**
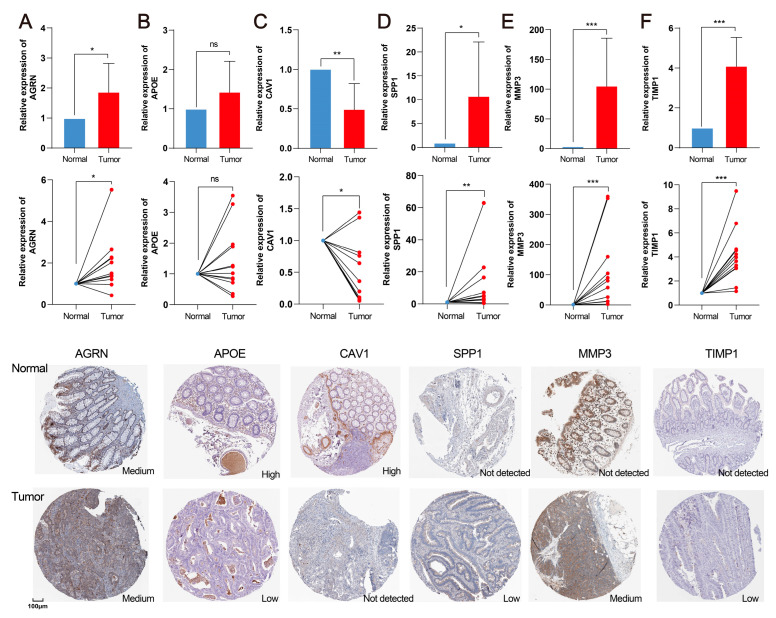
Validation of hub genes in the CRC clinical sample. Quantitative real-time polymerase chain reaction (qRT-PCR) analyses of AGRN (**A**), APOE (**B**), CAV1 (**C**), SPP1 (**D**), MMP3 (**E**), and TIMP1 (**F**) expression in 12 pairs of CRC tissues and adjacent non-cancer tissues. Immunohistochemistry showed the protein expressions of six hub genes based on the HPA database in normal and CRC tissues. ns *p* > 0.05. * *p* < 0.05, ** *p* < 0.001, and *** *p* < 0.001.

**Figure 10 ijms-24-13854-f010:**
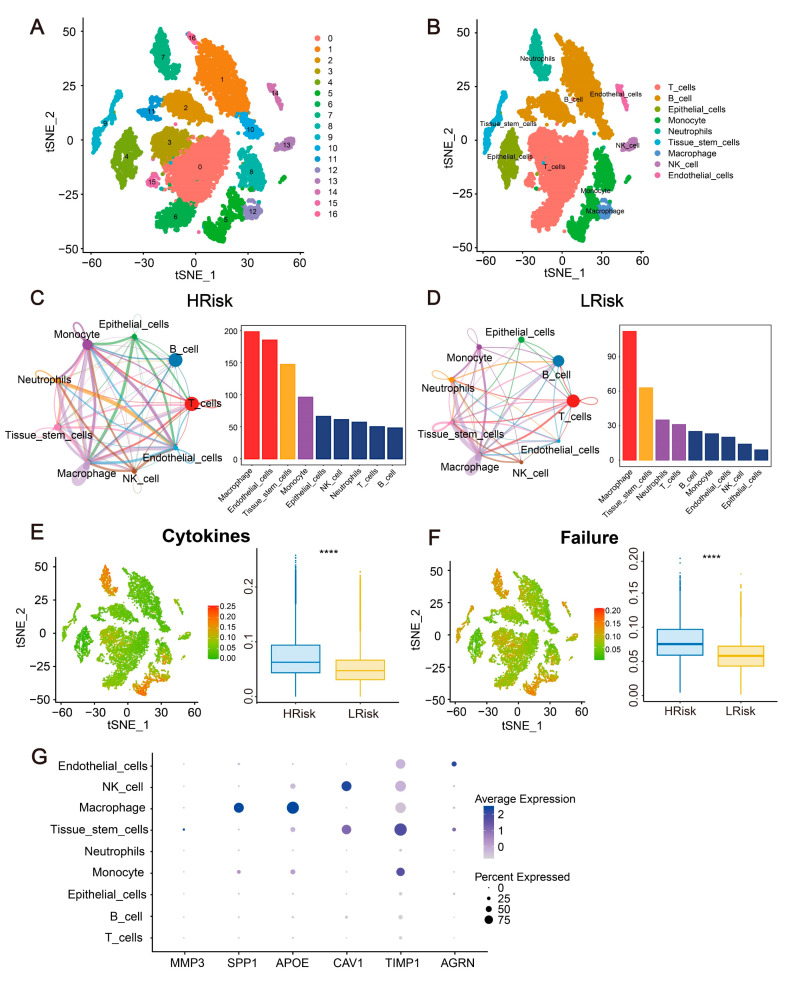
Single-cell analysis for the immune landscape. (**A**,**B**) A total of 9 cellular categories were identified from 17 subtypes. (**C**,**D**) Interaction patterns exist between these cellular subtypes in the high-risk and low-risk groups. (**E**,**F**) The failure scores and cytokine scores obtained were compared between the high- and low-risk groups. (**G**) The expression of hub genes in nine cell types. **** *p* < 0.0001.

## Data Availability

The bioinformatics data used in support of the findings of this study are public.

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
