# Peer review of "A Novel Lipid Metabolism and Endoplasmic Reticulum Stress-Related Risk Model for Predicting Immune Infiltration and Prognosis in Colorectal Cancer"

_ijms, 2023, doi:10.3390/ijms241813854_

Round 1
Reviewer 1 Report
In this paper, the Authors identified 18 differentially expressed genes related to lipid metabolism and endoplasmic reticulum (ER) stress.
The topic of the article is interesting, since the developed risk model can serve as a powerful prognostic tool, offering critical insights for refining clinical management and optimizing treatment in colorectal cancer (CRC) patients.
However, given the limitation of the study (lines 369-375), the impact of the research is too low to be published in IJMS.
Moreover, the description of the results and of the related figures lacks clarity because the molecular link between lipid metabolism alterations/ER stress and immune infiltration is not clear. Also the role of drug sensitivity in CRC patients remains cryptic.
In addition:
1) Lines 49-50: Sphingolipids are phospholipids or glycolipids. Therefore, the expression “Lipids, diverse in their classification, include phospholipids and sphingolipids” is not correct.
2) Lines 289-291: The text is repeated (but with different references!)
Reviewer 2 Report
Dear Editor, I reviewed the manuscript by Jin et al., entitled "A novel lipid metabolism and endoplasmic reticulum stress related risk model for predicting immune infiltration and prog nosis in colorectal cancer "
this manuscript discuss the the cross talk between Lipid metabolism and the ER stress as a predictive marker for colorectal cancer. In addition the manuscript is well written and the results are presented in best form and clear. Accordingly, the manuscript can be accepted for publication in the present form.
Many thanks
The english is fine, only minor editing is required
Reviewer 3 Report
The authors Jin H et al investigated the significance of lipid metabolism related genes and endoplasmic reticulum stress-related genes (LERGs) in colorectal cancer (CRC) using RNA sequencing and clinical data from databases to predict prognosis factors. The authors found the significant differences in immune cell infiltration and expression of immune-related factors between low risk and high risk CRC groups.
Minor comments:
In the abstract, conclusion include not only LERGs but also how lipid metabolism related genes modulates the disease.
Figure 8 G is difficult to see provide more clear images in the revised manuscript.
Reviewer 4 Report
Manuscript: 2548798
In this manuscript titled " A novel lipid metabolism and endoplasmic reticulum stress-related risk model for predicting immune infiltration and prognosis in colorectal cancer " by Jin et al., the authors aim to provide insights on lipid metabolism and endoplasmic reticulum stress-related gene signature correlation in CRC by using open access datasets. Though the study is comprehensive publicly available data mining and analysis, but it can gain by addressing the following concerns.
Authors claims that “Our prognostic model holds significant clinical value in both prognostic prediction and drug therapy”. Study uses a set of genes as hub genes and predict survival and drug therapy outcome based on risk score. Although the study can establish an association between gene signatures and CRC prognosis, the results will suffer from biological reproducibility and cannot enter clinical applications (PMID: 22846864, PMID: 33804045). It is very difficult to establish the biological validation using real world data for a set of genes to predict prognosis. For example, a clinician can investigate a biological marker to predict prognosis in real world but not the set of genes. See the studies below where in authors claims that a cell type or a gene can be independent prognosticator in lung cancer but not a set of genes.
“Circulating Giant Tumor-Macrophage Fusion Cells Are Independent Prognosticators in Patients With NSCLC” PMID: 32416323
“KIAA1522 is a novel prognostic biomarker in patients with non-small cell lung cancer” PMID: 27098511
I recommend authors to provide all the files of the manuscript if they are citing supplementary materials throughout the manuscript.
Please do not cite figures that are not provided in the manuscript. For example, figure 2 F and H.
I recommend authors to adhere to the standard of any scientific journals before they submit their study.
I recommend standard English proof reading of the manuscript and spelling and grammar check.
Round 2
Reviewer 1 Report
The quality of the manuscript is much improved as the result of the revision, and the authors have dealt with my comments and suggestion. As far as I am concerned, the manuscript is now acceptable to be published.